

**Large contribution of fossil-fuel derived secondary organic**
**carbon to water-soluble organic aerosols in winter haze of China**
Yan-Lin Zhang[1,2,3*], Imad El-Haddad[3], Ru-Jin Huang[3,4*], Kin-Fai Ho[4,5], Jun-Ji Cao[4*],
Yongming Han[4], Peter Zotter[3, #], Carlo Bozzetti[3], Kaspar R. Daellenbach[3], Jay G. Slowik[3], Gary
Salazar[2], André S.H. Prévôt[3*], Sönke Szidat[2*]
[1]Yale-NUIST Center on Atmospheric Environment, Nanjing University of Information Science
and Technology, 210044 Nanjing, China
[2]Department of Chemistry and Biochemistry & Oeschger Centre for Climate Change Research,
University of Bern, 3012 Bern, Switzerland
[3]Paul Scherrer Institute (PSI), 5232 Villigen, Switzerland
[4]Key Laboratory of Aerosol Chemistry and Physics, Institute of Earth Environment, Chinese
Academy of Sciences, 710061 Xi'an, China
[5]School of Public Health and Primary Care, The Chinese University of Hong Kong, Hong Kong,
China
[*]To whom correspondence should be addressed. E-mail: dryanlinzhang@outlook.com or
zhangyanlin@nuist.edu.cn (Y.-L.Z.); andre.prevot@psi.ch (A. Prévôt); rujin.huang@ieecas.cn
(R.-J.H.); jjcao@ieecas.cn (J.J.C.); szidat@dcb.unibe.ch (S.S.).
Phone: +86 25 5873 1022; fax: +86 25 5873 1193



**Abstract**
Water-soluble organic carbon (WSOC) is a large fraction of organic aerosols (OA) globally and
has significant impacts on climate and human health. The sources of WSOC remain very
uncertain in polluted regions.  Here we present a quantitative source apportionment of WSOC
isolated from aerosols in China using radiocarbon ($^{14}$C) and offline high-resolution time-of-
flight aerosol mass spectrometer measurements. Fossil emissions on average accounted for 32-
47% of WSOC. Secondary organic carbon (SOC) dominated both the non-fossil and fossil
derived WSOC, highlighting the importance of secondary formation to WSOC in severe winter
haze episodes. Contributions from fossil emissions to SOC were 61±4% and 50±9% in
Shanghai and Beijing, respectively, significantly larger than those in Guangzhou (36±9%) and
Xi'an (26±9%). The most important primary sources were biomass burning emissions,
contributing 17-26% of WSOC. The remaining primary sources such as coal combustion,
cooking and traffic were generally very small but not negligible contributors, as coal
combustion contribution could exceed 10%. Taken together with earlier $^{14}$C source
apportionment studies in urban, rural, semi-urban, and background regions in Asia, Europe and
USA, we demonstrated a dominant contribution of non-fossil emissions (i.e., 75±11%) to
WSOC aerosols in the North Hemisphere; however, the fossil fraction is substantially larger in
aerosols from East Asia and the East Asian pollution outflow especially during winter due to
increasing coal combustion. Inclusion of our findings can improve a modelling of effects of
WSOC aerosols on climate, atmospheric chemistry and public health.





## 1 INTRODUCTION

Water-soluble organic carbon (WSOC) is a large fraction of atmospheric organic

aerosols (OA), which contributes approximately 10% to 80% of the total mass of organic carbon
(OC) in aerosols from urban, rural and remote sites (Zappoli et al., 1999;Weber et al.,
2007;Ruellan and Cachier, 2001;Wozniak et al., 2012;Mayol-Bracero et al., 2002). Only 10 to
20% of total mass of WSOC has been resolved at a molecular level, and it consists of a large
variety of chemical species such as mono- and di-carboxylic acids, carbohydrate derivatives,
alcohols, aliphatic and aromatic acids and amino acids (Fu et al., 2015;Noziere et al., 2015).
Recent studies suggest that the water-soluble fraction of HUmic LIke Substances (HULIS) is
a major component of WSOC, which exhibits light-absorbing properties (Limbeck et al.,
2005;Andreae and Gelencser, 2006;Laskin et al., 2015). Therefore, WSOC has significant
influences on the Earth's climate either directly by scattering and absorbing radiation or
indirectly by altering the hygroscopic properties of aerosols and increasing cloud condensation
nuclei (CCN) activity (Asa-Awuku et al., 2011;Cheng et al., 2011;Hecobian et al., 2010).

WSOC can be directly emitted as primary particles mainly from biomass burning

emissions or produced from secondary organic aerosol (SOA) formation (Sannigrahi et al.,
2006;Kondo et al., 2007;Weber et al., 2007;Bozzetti et al., 2017b;Bozzetti et al., 2017a).
Ambient studies provide evidence that SOA formation through the oxidation of volatile organic
compounds (VOCs) and gas-to-particle conversion processes may be a prevalent source of
WSOC (Kondo et al., 2007;Weber et al., 2007;Miyazaki et al., 2006;Hecobian et al., 2010).
WSOC is therefore thought to be a good proxy of secondary organic carbon (SOC) in the
absence of biomass burning (Weber et al., 2007). By contrast, water-insoluble OC (WIOC) is
thought to be mainly from primary origins with a substantial contribution from fossil fuel
emissions (Miyazaki et al., 2006;Zhang et al., 2014b).

Due to a large variety of sources and unresolved formation processes of WSOC, their

relative fossil and non-fossil contributions are still poorly constrained. Radiocarbon ($^{14}$C)
analysis of sub-fractions of organic aerosols such as OC, WIOC and WSOC enable an





unambiguous, precise and quantitative determination of their fossil and non-fossil sources
(Zhang et al., 2012;Zhang et al., 2014b;Zhang et al., 2014c). Meanwhile, the application of
aerosol mass spectrometer measurement and positive matrix factorization and multi-linear
engine 2 (ME-2) can quantitatively classify organic aerosols into two major types such as
hydrocarbon-like OA (HOA) from primary fossil-fuel combustion and oxygenated organic
aerosol (OOA) from secondary origin (Zhang et al., 2007;Jimenez et al., 2009). Field
campaigns with the aerosol mass spectrometer (AMS) have revealed a predominance of OOA
in various atmospheric environments, although their sources remain poorly characterized
(Zhang et al., 2007;Jimenez et al., 2009). Previous studies found OOA is strongly correlated
with WSOC from urban aerosols in Tokyo, Japan, the Pearl River Delta (PRD) in South China
and Helsinki, Finland, indicating similar chemical characteristics, sources and formation
processes of OOA and WSOC (Kondo et al., 2007;Xiao et al., 2011;Timonen et al., 2013).
Similarly, HOA is mostly water insoluble and the major portion of water insoluble OC (WIOC)
can be assigned as HOA (Kondo et al., 2007;Daellenbach et al., 2016). Therefore, $^{14}$C
measurement of WIOC and WSOC aerosols may provide new insights into sources and
formation processes of primary and secondary OA, respectively, which also will elucidate the
origin of HOA and OOA as measured by AMS (Zotter et al., 2014b).
In this paper we apply a newly developed method to measure $^{14}$C in WSOC of PM$_{2.5}$
(particulate matter with an aerodynamic diameter of small than 2.5 μm) samples collected at
four Chinese megacities during an extremely severe haze episode during winter 2013 (Zhang
et al., 2015;Huang et al., 2014). In conjunction with our previous dataset from the same
campaign, we quantify fossil and non-fossil emissions from primary and secondary sources of
WSOC and WIOC. The dataset is also complemented by previous $^{14}$C-based source
apportionment studies conducted in urban, rural and remote regions in the North Hemisphere
to gain an overall picture of the sources of WSOC aerosols.
**2 MATERIALS AND METHODS**



### 2.1 Sampling

During January 2013 extremely high concentrations of 24-h $PM_{2.5}$ (i.e. often >100

$\mu g/m^3$) were identified in several large cities in East China (Huang et al., 2014;Zhang et al.,

2015). To investigate sources and formation mechanisms of the haze particles, an intensive

field campaign was carried out in four large cities, Beijing, Xi'an, Shanghai and Guangzhou,

which are representative cities of the Beijing-Tianjin-Hebei region, central-northwest region,

Yangtze Delta Region, and Pearl River Delta Region, respectively . The sampling procedures

have been previously described in detail elsewhere (Zhang et al., 2015). Briefly, $PM_{2.5}$ samples

were collected on pre-baked (450 °C for 6 hours) quartz filters using high-volume samplers for

24 h at a flow rate of ~1.05 $m^3$/min from 5 to 25 January 2013. The sampling sites in each city

were located at campuses of universities or at research institutes, at least 100 m away from

major emission sources (e.g., roadways, industry and domestic sources). One field blank sample

for each site was collected and analyzed. The results reported here were corrected for these field

blanks (Zotter et al., 2014a;Cao et al., 2013). All samples were stored at -20 °C before analysis.

The $PM_{2.5}$ mass was gravimetrically measured with an analytical microbalance before and after

sampling with the same conditions (~12 hour)

### 2.2 OC and EC mass determinations

A 1.0 $cm^2$ filter punches were used for OC and EC mass determination with a OC/EC

analyzer (Model4L) using the EUSAAR_2 protocol (Cavalli et al., 2010). The replicate analysis

(n=6) showed an analytical precision with relative standard deviations smaller than 5%, 10%,

and 5% for OC, EC and TC, respectively. The field blank of OC was on average 2.0 ± 1.0

$\mu g/cm^2$ (equivalent to ~0.5 $\mu g/m^3$), which was used for blank correction for OC. EC data was

not corrected for field blank, because such a blank was not detectable.

### 2.3 Offline-AMS measurement and PMF source apportionment

The water-soluble extracts from the same samples were analyzed by a high resolution time of

flight aerosol mass spectrometer (HR-ToF-AMS) and the resulting mass spectra were used as





an inputs for positive matrix factorization (PMF) for the source apportionment of the WSOC,
OC and PM$_{2.5}$. The methodology applied, and the AMS-PMF results obtained are detailed in
Huang et al. (2014) and will only be briefly described in the following.
Filter punches (the equivalent of ~4 cm$^2$) were sonicated in 10 mL ultrapure water (18.2 MΩ
cm at 25 °C, TOC <3ppb) for 20 min at 30°C. The water extracts were aerosolized and the
resulting particles were dried with a silica gel diffusion dryer before analysis by the HR-ToF-
AMS. For each measurement ten mass spectra were recorded (AMS V-mode, m/z 12-500), with
a collection time for each spectrum of 1 minute.
Online AMS measurements provide quantitative mass spectra of submicron non-refractory
aerosol species, including organic aerosol and ammonium nitrate and sulfate. However, the
offline AMS measurements described herein cannot be directly related to ambient
concentrations due to uncertainties in filter extraction and nebulization. In Huang et al. (2014),
the obtained mass spectra were scaled to the organic aerosol mass, obtained from the organic
carbon concentrations from the Sunset analyzer times the OM/OC ratios determined by the
AMS. Since the AMS-measured species are not extracted and aerosolized with equal efficiency,
PMF outputs were corrected using factor dependent recoveries. Here, this correction, which is
the main source of uncertainties reported in Huang et al. (2014), was not required any more
since here only WSOC PMF is present and organic aerosol mass spectra are directly scaled to
water soluble organic aerosol concentrations (WSOM, obtained as WSOC times OM/OC ratios).
The quantitative WSOM mass spectra are used together with other aerosol species (listed
below), collectively referred to as 'species' hereafter, as PMF inputs. PMF solves the bilinear
matrix equation:
$$X_{ij} = \sum_k G_{i,k} F_{k,j} + E_{i,j} \qquad \text{(Eq. 1)}$$
by following the weighted least squares approach. In the equation, $i$ represent the time index, $j$
a species and $k$ the factor number. $X_{ij}$ is the input matrix and $s_{i,j}$ the corresponding error matrix.
$G_{i,k}$ is the matrix of the factor time-series, $F_{k,j}$ is the matrix of the factor profiles and $E_{i,j}$ the



model residual matrix. PMF determines $G_{i,k}$ and $F_{k,j}$ such that the ratio of the Frobenius norm
of $E_{i,j}$ over $s_{i,j}$ is minimised.
The species considered as inputs include the quantitative WSOM mass spectra, organic markers
(3 anhydrous sugars, 4 lignin breakdown products, 2 resin acids, 4 hopanes, 19 polycyclic
aromatic hydrocarbons and their oxygenated derivatives), EC, and major ions (Cl$^-$, NO$_3^-$, SO$_4^{2-}$,
oxalate, methylsulfonic acid, Na$^+$, K$^+$, Mg$^{2+}$, Ca$^{2+}$, and NH$_4^+$) and residual PM. The latter is the
difference between total PM$_{2.5}$ mass and the measured species. It represents our best estimate
of the particulate chemical species not measured here, most likely dominated by crustal material.
The Source Finder toolkit (SoFi v.4.9) (Canonaco et al., 2013) for IGOR Pro software package
(Wavemetrics, Inc., Portland, OR, USA) was used to run the PMF algorithm. The PMF was
solved by the Multilinear Engine 2 (ME-2, Paatero, 1999), which allows the constraining of the
$F_{k,j}$ elements to vary within a certain range defined by the scalar α ($0 \leq \alpha \leq 1$), such that the
modelled $F'_{k,j}$ equals:
$$F'_{k,j} = F_{k,j} \pm \alpha * F_{k,j} \qquad\qquad\qquad \text{(Eq. 2)}$$
The elements that were constrained in $F_{k,j}$ matrix can be found in Huang et al. (2014). The
factors extracted by ME-2 were interpreted to be related to primary emissions from traffic (TR),
biomass burning (BB), coal burning (CC), cooking emissions (CI) and dust and from two
secondary aerosol fractions. The contribution of the water soluble organic aerosol related to
these different factors are extracted and divided by the respective OM/OC$_k$ calculated from the
high-resolution analysis of the factor mass spectral profile, to obtain the WSOC$_k$ time series
related to each of the factors. In the following analysis, the mass of WSOC$_k$ related to coal
burning and traffic were assigned to fossil WSOC fraction, while the mass of WSOC$_k$ related
to biomass burning and cooking emissions were assigned to non-fossil WSOC fraction (see Sec.
2.5). Meanwhile, the remaining WSOC fractions are assigned to the secondary factors, which
can be from both fossil and non-fossil origins, were considered collectively and compared to



the unassigned fossil and non-fossil WSOC, to retrieve the origins of this remaining fraction
(see Sec. 2.5).
**2.4 $^{14}$C measurement of WSOC**

$^{14}$C content of micro-scale WSOC aerosol samples was measured with a newly

developed method (Zhang et al., 2014c). Briefly, a 16-mm-diameter punch of each filter was
extracted using 10 ml ultrapure water with low TOC impurity (less than 5 ppb). The water
extracts were recovered in the 20 ml PFA vials and were then pre-frozen at -20 °C more than 5
hours before completely dryness in a freeze dryer (Alpha 2-4 LSC, Christ, Germany) for about
24 h to 36 h. The residue was re-dissolved in 50 μl of ultrapure water three times and transferred
into 200 μl tin capsules (Elementar, Germany). The concentrated samples were heated in the
oven at 55-60 °C until complete dryness before the $^{14}$C measurements.

WSOC extracts in tin capsules were then converted to $CO_2$ by the oxidation of the

carbon-containing samples using an Elemental Analyzer (EA, Model Vario Micro, Elementar,
Germany) as a combustion unit (up to 1050 °C). The resulting $CO_2$ was introduced continuously
by a versatile gas inlet system into a gas ion source of the accelerator mass spectrometer
MICADAS where $^{14}$C of $CO_2$ was finally measured (Wacker et al., 2013;Salazar et al., 2015).
The $^{14}$C content of OC and EC was measured in our previous study (Zhang et al., 2015). $^{14}$C
results were expressed as fraction of modern ($f_M$), i.e., the fraction of the measured $^{14}$C/$^{12}$C ratio
related to the $^{14}$C/$^{12}$C ratio of the reference year 1950 (Stuiver, 1977). To correct excess $^{14}$C
from nuclear bomb tests in the 1950s and 1960s, $f_M$ values were converted to the fraction of
non-fossil ($f_{NF}$) (Zotter et al., 2014a;Zhang et al., 2012):

$f_{NF} = f_M / f_{M,ref}$ (Eq. 3)

$f_{M,ref}$ is a reference value of $f_M$ for non-fossil carbon sources including biogenic and

biomass burning emissions, which were estimated as 1.08±0.05 for WSOC samples collected
in 2013 according to the contemporary atmospheric $CO_2$ $f_M$ (Levin et al., 2010) and a tree
growth model (Mohn et al., 2008).





### 2.5 AMS²-based source apportionment of WSOC

To better understand the origin of WSOC observed at these sites, WSOC sources were apportioned into several major sources by a combination of $^{14}$C and PMF source apportionments (See Figure 1). Here, two "AMS" (i.e., accelerator mass spectrometer and aerosol mass spectrometer), such a combined approach was named as "AMS²-based source apportionment.

WSOC concentration from non-fossil ($WSOC_{NF}$) and fossil ($WSOC_F$) sources were calculated from:

$$WSOC_{NF}=WSOC*f_{NF}(WSOC) \quad \text{(Eq. 4)}$$

$$WSOC_F=WSOC-WSOC_{NF} \quad \text{(Eq. 5)}$$

The mass concentration of WSOC was derived from the subtraction of TC mass measured from a water-extracted filter from that measured with an un-treated filter (Zhang et al., 2012):

$$WSOC=TC_{un\text{-}treated}-TC_{water\text{-}extracted} \quad \text{(Eq.6)}$$

Based on mass balance, WIOC concentrations from non-fossil ($WIOC_{NF}$) and fossil ($WIOC_F$) sources were calculated from:

$$WIOC_{NF}=OC_{NF}-WSOC_{NF} \quad \text{(Eq. 7)}$$

$$WIOC_F=OC_F-WSOC_F \quad \text{(Eq.8)}$$

where OC concentrations from non-fossil ($OC_{NF}$) and fossil ($OC_F$) sources were obtained by mass and $^{14}$C measurement of the OC fraction, which were reported previously (Zhang et al., 2015).

The non-fossil and fossil-fuel derived WSOC can be apportioned into primary and secondary OC:



$WSOC_{NF}=WSOC_{POC,NF}+WSOC_{SOC,NF}$ (Eq.9)

$WSOC_{F}= WSOC_{POC,F}+WSOC_{SOC,F}$ (Eq.10)

$WSOC_{POC,NF}$ can be sub-divided into the following three major primary emissions including
cooking emission ($WSOC_{CI}$) and biomass burning ($WSOC_{BB}$).

$WSOC_{POC,NF}=WSOC_{CI}+WSOC_{BB}$ (Eq.11)

Similarly, $WSOC_{POC,F}$ can be sub-divided into the following two major primary emissions
including traffic ($WSOC_{TR}$) and coal combustion ($WSOC_{CB}$).

$WSOC_{POC,F}=WSOC_{TR}+WSOC_{CB}$ (Eq.12)

where primary fractions such as $WSOC_{CI}$, $WSOC_{BB}$, $WSOC_{TR}$ and $WSOC_{CB}$ are
previously estimated by the off-line AMS-PMF approach (Huang et al., 2014;Daellenbach et
al., 2016;Bozzetti et al., 2017a;Bozzetti et al., 2017b).
An uncertainty propagation scheme using a Latin-hypercube sampling (LHS) model
was implemented to properly estimate overall uncertainties including measurement
uncertainties of the mass determinations of carbon species (i.e., OC, EC, TC, WSOC, WIOC)
and $^{14}C$ measurement, blank corrections from field blanks, and estimation of $f_{M,ref}$ (Zhang et al.,

2015).

**3 RESULTS AND DISCUSSION**
**3.1 Overall results**
During the haze periods of January 2013, the highest daily average $PM_{2.5}$ concentrations were
found in Xi'an (345 µg/m$^3$) followed by Beijing (158 µg/m$^3$), Shanghai (90 µg/m$^3$) and
Guangzhou (68 µg/m$^3$). These levels were much higher than the China's National ambient Air
quality standards (i.e., 35 µg/m$^3$). Indeed, several studies have already reported the chemical
composition, source and formation mechanism of $PM_{2.5}$ in many large cities during the haze
events of January 2013 in East China. For examples, Huang et al. (2014) revealed that the



secondary aerosol formation contributed to 44–71% of OA in Beijing, Xi'an, Shanghai, and
Guangzhou during this extremely haze event in China (Huang et al., 2014). By $^{14}$C-based source
appointment conducted in the same campaign, Zhang et al. (2015) have reported that
carbonaceous aerosol pollution was driven to a large (often dominant) extent by SOA formation
from both, fossil and biomass-burning sources (Zhang et al., 2015). For all four cities, the 24 h
average levels of WSOC were significantly correlated with the levels of $PM_{2.5}$ and OC ($R$=0.99,
$p$<0.01, Figure 2), suggesting that WSOC and OA may have similar sources and formation
processes and thus have important implications for OC loadings and associated environmental
and health effects. However, the sources of WSOC remain poorly constrained. In this study,
we measured the $^{14}$C content of WSOC aerosols in six samples (three with the highest three
with average PM mass) for each city to report on heavily and moderately polluted days (HPD
and MPD, respectively) (Zhang et al., 2015). The $^{14}$C contents of OC and EC of the same
samples were reported previously (Zhang et al., 2015).
WSOC on average accounted for 53±8.0 (ranging from 40-65%) of OC including all samples
from the four sites, which was consistent with previous estimates . Based on these
measurements, the concentrations of WSOC from non-fossil sources ($WSOC_{NF}$) spanned from
1.41 to 45.3 μg/m$^3$ with a mean of 10.6±12.1 μg/m$^3$, whereas the corresponding range for
WSOC from fossil-fuel emissions ($WSOC_F$) was 0.44 to 20.1 μg/m$^3$ with a mean of 5.3±4.9
μg/m$^3$ (Figure 3). Similar to $PM_{2.5}$ levels, the highest concentrations of $WSOC_{NF}$ and $WSOC_F$
were observed in Northern China in Xi'an and Beijing (Xi'an>Beijing), followed by the two
southern sites Shanghai and Guangzhou. Non-fossil contributions (mean ± standard deviation)
to total WSOC were 53±5%, 75±4%, 48±2% and 68±6% in Beijing, Xi'an, Shanghai, and
Guangzhou, respectively. Thus, fossil contributions were notably higher in Beijing and
Shanghai than those in Xi'an and Guangzhou. Such a trend was also observed for OC (Zhang
et al., 2015), suggesting relatively high contribution from fossil-fuel emissions to OC and
WSOC due to large coal usage. Despite of these fossil emissions, non-fossil sources were
considerably important or even dominant contributors for all the studied sites, which may be



associated with primary and secondary OA from regional-transported and local biomass
burning emissions. It should also be noted that formation of SOA derived from biogenic VOCs
may also have contributed to WSOC$_{NF}$ in Guangzhou, where temperatures during the sampling
period were significantly higher (i.e., 5–18 °C) than those in other cities (i.e., -12 to
+9 °C)(Bozzetti et al., 2017b). Although both fossil and non-fossil WSOC concentrations were
dramatically enhanced during HPD compared to those during MPD, their relative contributions
did not change significantly in Beijing and Shanghai whereas a small increasing and decreasing
trend in non-fossil fraction was found in Xi'an and Guangzhou, respectively (Figure 3). This
suggests that the source pattern of WSOC in Beijing and Shanghai remained similar between
HPD and MPD, but the increase in the WSOC concentrations was rather enhanced by additional
fossil-fuel and biogenic/biomass burning emissions in Guangzhou and Xi'an, respectively. It
should be noted that the meteorological conditions play significant roles on the haze formation
in the eastern China during winter 2013, and has already been well documented (Zhang et al.,
2014a). However, the details sources of WSOC and WIOC were still unclear.

**3.2 WSOC versus WIOC**

To compare sources of WSOC and WIOC aerosols, the mass concentrations and $^{14}$C contents
of WIOC were also derived based on mass balance. The $^{14}$C-based source apportionment of
WIOC and the relationship between $f_{NF}$(WSOC) and $f_{NF}$(WIOC) is presented in Figures 4 and
5a, respectively. It shows that non-fossil contributions to WSOC were larger than those of
WIOC for nearly all samples in Beijing, Xi'an and Guangzhou. On average, the majority (60-
70%) of the fossil OC was water insoluble at these 3 sites (see Figure 5b), indicating that fossil-
derived OA mostly consisted of hydrophobic components and thus is less water soluble than
OA from non-fossil sources. This result is consistent with findings reported elsewhere such as
at an urban or rural site in Switzerland (Zhang et al., 2013), a remote site in Hainan Island,
South China (Zhang et al., 2014b) and at two rural sites on the east coast of the United States
(Wozniak et al., 2012). Meanwhile, the fossil OC in Shanghai, the dominant fraction of OC,
was more water soluble (Figure 5b), suggesting an enhanced SOA formation from fossil VOCs


from vehicle emissions and/or coal burning for this city. As shown in Figure 5b, non-fossil OA
was enriched in water-soluble fractions (i.e., 60%±8%) for all cities, associated with the
hydrophilic properties of biogenic-derived SOA and biomass-burning derived primary organic
aerosol (POA) and SOA, which are composed of a large fraction of polar and highly oxygenated
compounds (Mayol-Bracero et al., 2002;Sullivan et al., 2011;Noziere et al., 2015). Thus, non-
fossil OC has more water-soluble components than fossil ones. It should be noted that relative
contributions of $WSOC_{NF}$ and $WSOC_F$ are similar in Beijing and Shanghai, whereas $WSOC_{NF}$
is much higher than $WSOC_F$ in Xi'an and Guangzhou. This suggests larger contribution of non-
fossil sources to WSOC aerosols in Xi'an and Guangzhou than those in Beijing and Shanghai.
**3.3 High contribution of secondary formation to WSOC**
WSOC was further apportioned into fossil sources such as coal burning (CB), traffic (TR) and
SOC (SOC,F) as well as non-fossil sources such as biomass burning (BB), cooking (CI) and
SOC (SOC,NF) using a $AMS^2$ based source apportionment (see Sec. 2.5 and Figure 1). SOC
dominated WSOC during both the HPD and MPD with a mean contribution of 67%±9%,
highlighting the importance of SOC formation to the WSOC aerosols in wintertime pollution
events. This is consistent with our previous findings for total $PM_{2.5}$ mass and bulk carbonaceous
aerosols (i.e., total carbon, sum of OC and EC) (Huang et al., 2014;Zhang et al., 2015). The
increase in SOC contribution to WSOC during HPD compared to MPD can be largely due to
fossil contribution in Beijing but non-fossil emissions in Xi'an. In Shanghai and Guangzhou,
the source pattern of WSOC was not significantly different between MPD and HPD. Fossil
contributions to $WSOC_{SOC}$ were 50%±9% in Beijing, 61±4% in Shanghai, associated with SOA
from local and transported fossil-fuel derived precursors at these sites (Guo et al., 2014). This
contribution drops to 36±9% and 26±9% in Guangzhou and Xi'an, respectively, due to higher
biomass-burning contribution to SOC. Despite of the general importance of fossil SOC,
formation of non-fossil $WSOC_{SOC}$ becomes especially relevant during HPD especially in Xi'an
(Figure 6), which may be explained by competing effects in SOC formation from fossil versus
non-fossil precursors. It can be hypothesized for extremely polluted episodes that more



hydrophilic volatile compounds that were emitted from biomass burning precursors
preferentially form SOC compounds via heterogeneous reaction/processing on dust particles
compared to highly hydrophobic precursors from fossil sources, a point subjected to future
laboratory and field experiments. The most important primary sources of WSOC were biomass
burning emissions, and their contributions were higher in Xi'an (26%±7%) and Guangzhou
(25%±6%) than those found in Beijing (17%±6%) and Shanghai (17%±5%). The remaining
primary sources such as coal combustion, cooking and traffic were generally very small
contributors of WSOC due to lower water solubility, although coal combustion could exceed
10% in Beijing. It should be noted that WSOC was dominated by SOC formation with mean
contribution of 61%±10% and 72%±12% (average for all four cities) to non-fossil and fossil-
fuel derived WSOC, respectively.

**Summary and implications**

Our study demonstrates that non-fossil emissions are generally a dominant contributor of
WSOC aerosols during extreme haze events in representative major cities of China, which is in
agreement with WSOC source information identified in aerosols with different size fractions
(e.g., TSP, $PM_{10}$ and $PM_{2.5}$) observed in the Northern Hemisphere at urban, rural, semi-urban,
and background sites in East/South Asia, Europe and USA (Table 1). The $^{14}$C-based source
apportionment database shows a mean non-fossil fraction of 73±11% across all sites. This
overwhelming non-fossil contribution to WSOC is consistently observed throughout the year,
which is associated with seasonal-dependent biomass-burning emissions and/or biogenic-
derived SOC formation. Our study provides evidence that the presence of oxidized OA, which
is to a large extent water soluble, in the Northern Hemisphere (Zhang et al., 2007) is mainly
derived from biogenic-derived SOA and/or biomass burning sources. The overall importance
of non-fossil emissions to the WSOC aerosols results from large contributions of SOC
formation from biogenic precursors (e.g., most likely during summer) and relatively high water-
solubility of primary biomass burning particles (e.g., most likely during winter) compared to
those emitted from fossil fuel emissions such as coal combustion and vehicle exhaust. Despite



of the importance of non-fossil sources, a significant fossil fraction is also observed in the
WSOC aerosols from polluted regions in East Asia and sites influenced by East Asian
continental outflow (Table 1, Figure 7). This fossil contribution is apparently higher than in this
region than in the USA and Europe, which is due to large industrial and residential coal usage
as well as vehicle emissions. From our observation, the increases in the fossil fractions of
WSOC were mostly from SOC formation. Since WSOC has hygroscopic properties, our
findings suggest that SOC formation from non-fossil emissions have significant implications
on aerosol-induced climate effects. In addition, fossil-derived SOC formation may also become
important in polluted regions with large amounts of fossil fuel emissions such as in China and
other emerging countries. Low combustion efficiencies and consequently high emission factors
in most of the combustion processes in China may further be responsible for increased
concentrations of fossil precursors which may be oxidized to form water-soluble SOA in the
atmosphere and contribute substantially to the WSOC aerosols. The enhanced WSOC levels
may be also originate from aging of fossil POA during the long-range transport of aerosols
(Kirillova et al., 2014a). It is also interesting to note that fossil contribution during winter in
East Asia is generally higher than those in the rest of the year although relatively large fossil
fraction could be occasionally found as well. Such seasonal dependence was not observed in
other regions, suggesting the importance of fossil contribution to WSOC due to increasing coal
combustions during winter in China. This study provides a more detailed source apportionment
of WSOC, which could improve modelling of climate and health effects as well as the
understanding of atmospheric chemistry of WSOC in the polluted atmosphere such as China
and provide scientific basis for policy decisions on air pollution emissions mitigation.

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

**Author Contributions:** Y.-L.Z., S.S., R.J.H., J.J.C. and A.S.H.P. designed the study. Y.L.Z.
and G.S. perform $^{14}$C measurement. Y.L.Z. and S. S. interpreted the $^{14}$C data. R.J.H., I.E.H.,
C.B. and K.D. performed the offline AMS analysis and interpret the data. Y.-L.Z. and I.E.H.
perform $^{14}$C-AMS-PMF source apportionments. Y.-L.Z. wrote the paper. All authors reviewed
and commented on the paper.
**Competing interests:** The authors declare no competing financial interests.
**Acknowledgments:** This work was supported by the National Natural Science Foundation of
China (Grant Nos. 91644103, 41603104). All data needed to evaluate the conclusions in the
paper are present in the paper. Additional data related to this paper may be requested from the
authors.





**Figures and Tables**
**Table 1.** Compilation of literature values of relative fossil-fuel contributions (fossil %) to the
WSOC aerosols in East/South Asia, USA and Europe.

| Site | Location | Season | Size | WSOC (µg/m³) | WSOC/OC | Fossil % | References |
|---|---|---|---|---|---|---|---|
| **East Asia** | | | | | | | |
| Urban | Beijing, China | Winter/2013 | PM$_{2.5}$ | 19.8 | 0.49 | 47 | this work |
| Urban | Xi'an, China | Winter/2013 | PM$_{2.5}$ | 31.3 | 0.53 | 25 | this work |
| Urban | Shanghai, China | Winter/2013 | PM$_{2.5}$ | 6.5 | 0.58 | 52 | this work |
| Urban | Guangzhou, China | Winter/2013 | PM$_{2.5}$ | 6.6 | 0.53 | 32 | this work |
| Urban | Beijing, China | Winter/2014 | PM$_{2.5}$ | 14.7 | 0.40 | 56 | (Fang et al., 2017) |
| Urban | Beijing, China | Winter/2011 | PM$_{4.3}$ | 15 | 0.50 | 55 | (Zhang et al., 2014c) |
| Urban | Beijing, China | Winter/2013 | PM$_{2.5}$ | 9.3 | 0.31 | 54 | (Yan et al., 2017) |
| Urban | Guangzhou, China | Winter/2012/2013 | PM$_{2.5}$ | 4.1 | 0.38 | 33 | (Liu et al., 2014) |
| Urban | Guangzhou, China | Winter/2011 | PM$_{10}$ | 4.5 | 0.43 | 28.5 | (Zhang et al., 2014c) |
| Urban | Xi'an, China | Autumn/2009 | PM$_{2.5}$ | 5.1 | 0.28 | 31 | (Pavuluri et al., 2013) |
| Urban | Xi'an, China | Autumn/2010 | TSP | 8.1 | 0.28 | 29 | (Pavuluri et al., 2013) |
| Urban | Wuhan, China | Winter/2013 | PM$_{2.5}$ | 13.7 | 0.45 | 37 | (Liu et al., 2016) |
| Urban | Sapporo, Japan | Summer/Autumn/2010 | PM$_3$ | 1 | 0.43 | 15 | (Pavuluri et al., 2013) |
| Urban | Sapporo, Japan | Summer/2011 | TSP | 1.1 | 0.24 | 12 | (Pavuluri et al., 2013) |
| Urban | Sapporo, Japan | Spring/2010 | TSP | 1.1 | 0.31 | 11 | (Pavuluri et al., 2013) |
| Urban | Sapporo, Japan | Autumn/2011 | TSP | 1.8 | 0.48 | 18.3 | (Pavuluri et al., 2013) |
| Urban | Sapporo, Japan | Winter/2010 | TSP | 0.9 | 0.45 | 40.2 | (Pavuluri et al., 2013) |
| Background | Jeju Island, Korea | Winter/2014 | PM$_{2.5}$ | 2.2 | 0.66 | 50 | *(Fang et al., 2017)* |
| Background | Jeju Island, Korea | Spring/2011 | PM$_{2.5}$ | 2.0 | | 37.5 | *(Kirillova et al., 2014a)* |
| Background | Jeju Island, Korea | Spring/2011 | TSP | 3.0 | | 25 | *(Kirillova et al., 2014a)* |
| **Average** | | | | | | 33±14 | |
| **South Asia** | | | | | | | |
| Background | Hainan, China | Annual 2005/2006 | PM$_{2.5}$ | 3.9 | 0.54 | 18 | (Zhang et al., 2014b) |
| Background | Hainan, China | Winter 2005/2006 | PM$_{2.5}$ | 6.2 | 0.57 | 14.5 | (Zhang et al., 2014b) |
| Background | Hainan, China | Summer 2005/2006 | PM$_{2.5}$ | 1.4 | 0.40 | 17.7 | (Zhang et al., 2014b) |
| Background | Hanimaadhoo, Maldives | Annual 2008/2009 | TSP | 0.5 | | 17 | (Kirillova et al., 2013) |
| Background | Sinhagad, India | Annual 2008/2009 | TSP | 3.0 | | 24 | (Kirillova et al., 2013) |
| Background | Hanimaadhoo, Maldives | Spring/2012 | PM$_{2.5}$ | 0.6 | 0.62 | 14 | (Bosch et al., 2014) |
| Urban | Delhi, India | Winter/2010/2011 | PM$_{2.5}$ | 22.0 | | 21 | (Kirillova et al., 2014b) |
| Average | | | | | | 18±4 | |



| **Europe and USA** | | | | | | | |
|---|---|---|---|---|---|---|---|
| Urban | Göteborg, Sweden | Winter/2005 | PM$_{2.5}$ | 1.1 | 0.48 | 23 | (Szidat et al., 2009) |
| Urban | Göteborg, Sweden | Summer/2006 | PM$_{2.5}$ | 0.8 | 0.61 | 30 | (Szidat et al., 2009) |
| Rural | Göteborg, Sweden | Winter/2005 | | 1.2 | 0.53 | 27 | (Szidat et al., 2009) |
| Rural/semi-urban | Stockholm, Sweden | Summer/2009 | TSP | | | 12 | (Kirillova et al., 2010) |
| Urban | Zürich, Switzerland | Summer/2002 | PM$_{10}$ | 2.1 | 0.54 | 14 | (Szidat et al., 2004) |
| Urban | Zürich, Switzerland | Winter/2008 | PM$_{10}$ | 2.8 | 0.60 | 26.8 | (Zhang et al., 2013) |
| Urban | Moleno, Switzerland | Summer/2006 | PM$_{10}$ | 5.3 | 0.67 | 30 | (Zhang et al., 2013) |
| Urban | Bern, Switzerland | Winter/2009 | PM$_{10}$ | | 0.39 | 14 | (Zhang et al., 2014c) |
| Urban | Atlanta, USA | Summer/2004 | PM$_{2.5}$ | 2.3 | 0.59 | 26.5 | (Weber et al., 2007) |
| Rural | Millbrook, USA | Annual/2006/2007 | TSP | | 0.36 | 12 | (Wozniak et al., 2012) |
| Rural | Harcum, USA | Annual/2006/2007 | TSP | | 0.38 | 14 | (Wozniak et al., 2012) |
| **Average** | | | | | | 21±8 | |


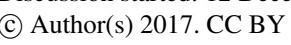



**Figure 1**. The AMS[2]-based source apportionment scheme of WSOC aerosols in this study.
See the main text for the equations (i.e., Eq. 4, 5, 9, 10 in the Sec. 2.5) and the offline-AMS &
PMF (see the Sec. 2.3).

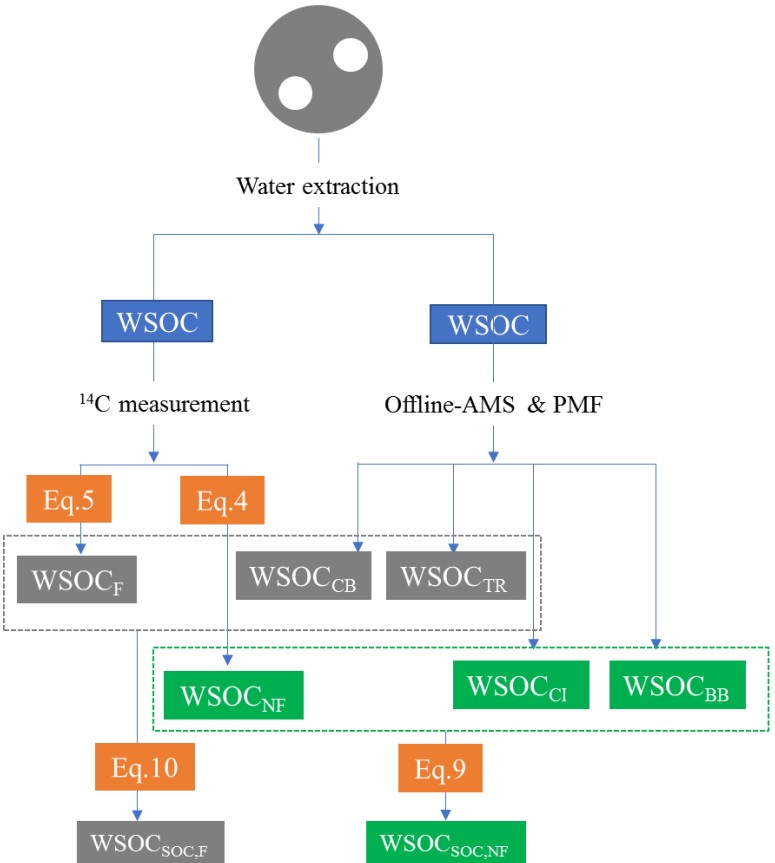




**Figure 2**. Linear relationships (p<0.01) of WSOC with PM$_{2.5}$ (top) and OC concentrations
(bottom).

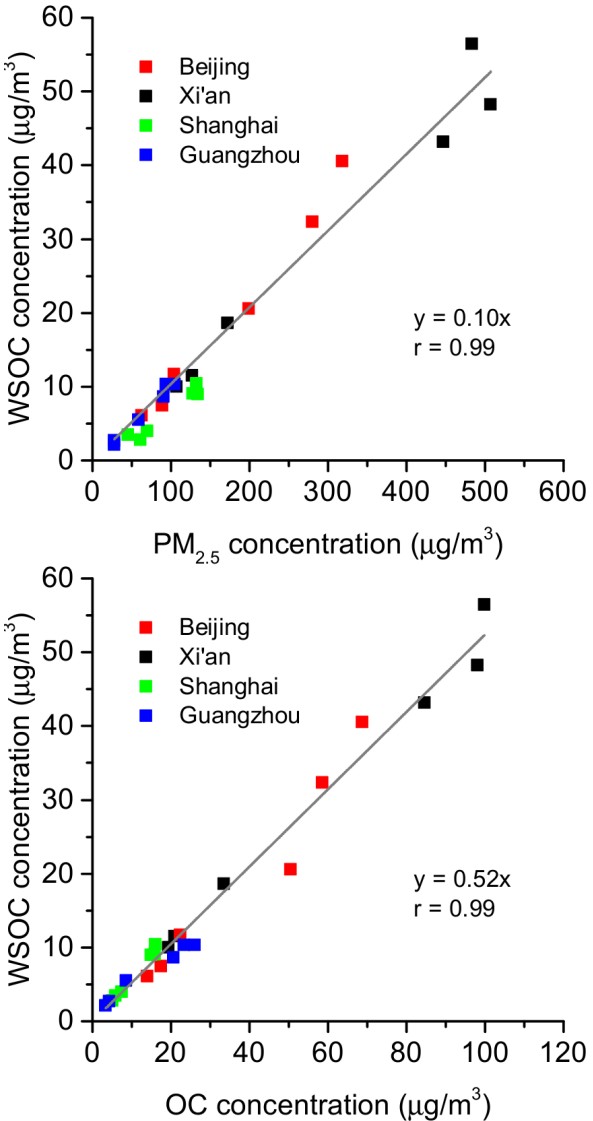




**Figure 3**. Mass concentrations (μg/m³) of WSOC from non-fossil and fossil-fuel sources
(WSOC$_{NF}$ and WSOC$_F$, respectively) as well as non-fossil fractions of the WSOC aerosols from
Beijing, Xi'an, Shanghai and Guangzhou during moderately polluted days (MPD) and heavily
polluted days (HPD). Note the different scaling for different cities.

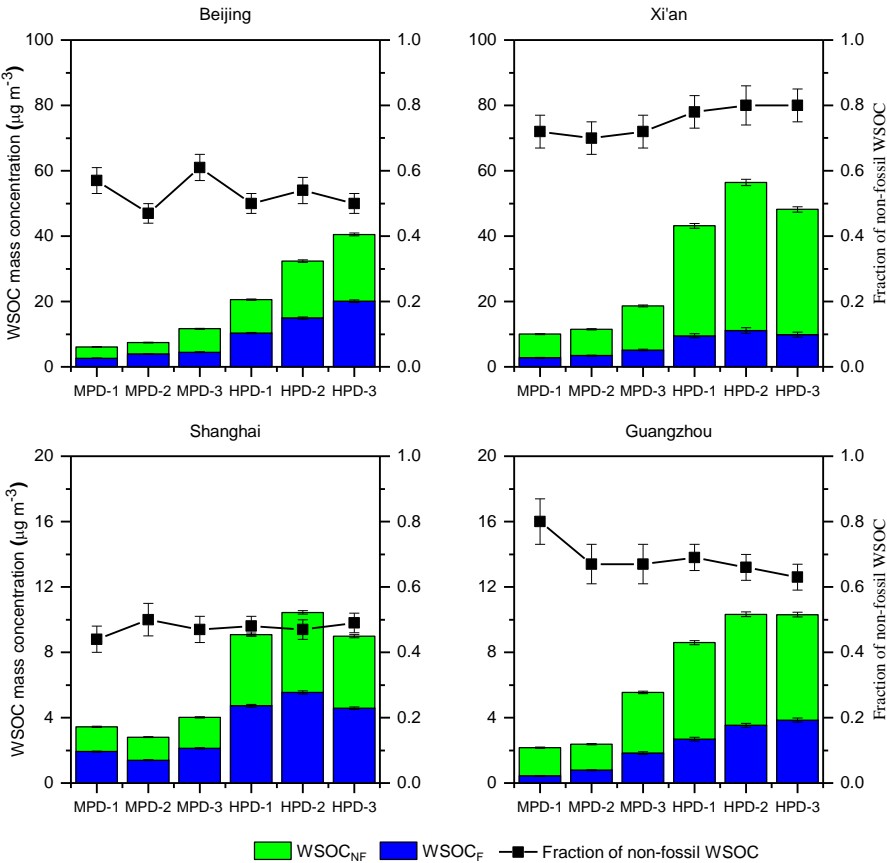




**Figure 4**. Mass concentrations (µg/m³) of WIOC from non-fossil and fossil-fuel sources
(WIOC NF and WIOC F, respectively) as well as non-fossil fractions in the WIOC aerosols from
Beijing, Xi'an, Shanghai and Guangzhou during moderately polluted days (MPD) and heavily
polluted days (HPD). Note the different scaling for different cities.

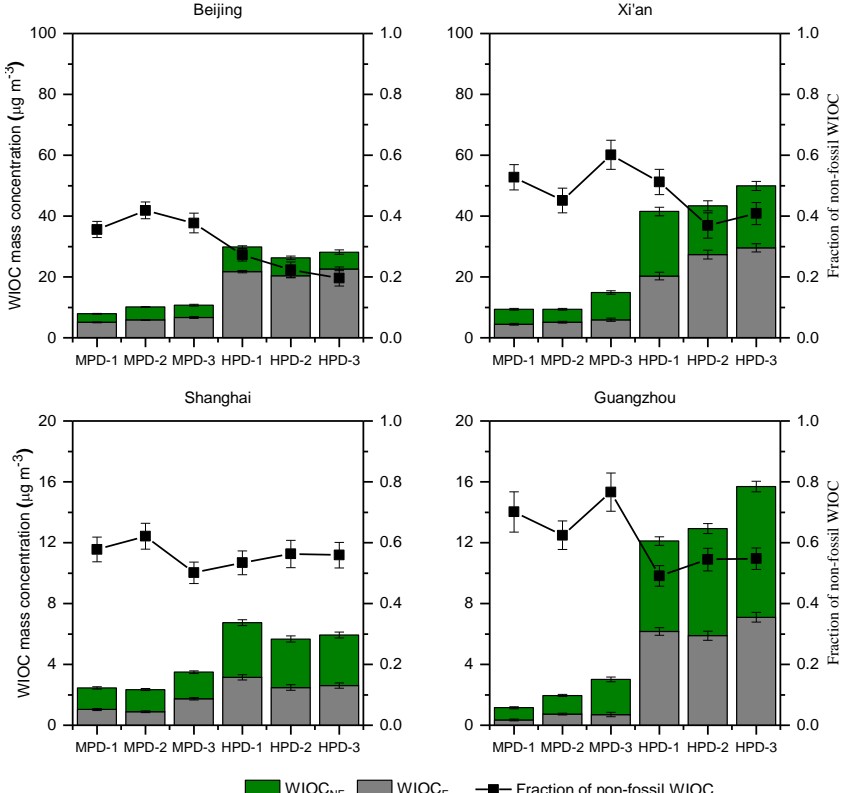




**Figure 5**. Relationship between the fraction of non-fossil WIOC and WSOC(a) and averaged
relative contribution (%) to OC from WSOC and WIOC from non-fossil and fossil sources
(b).

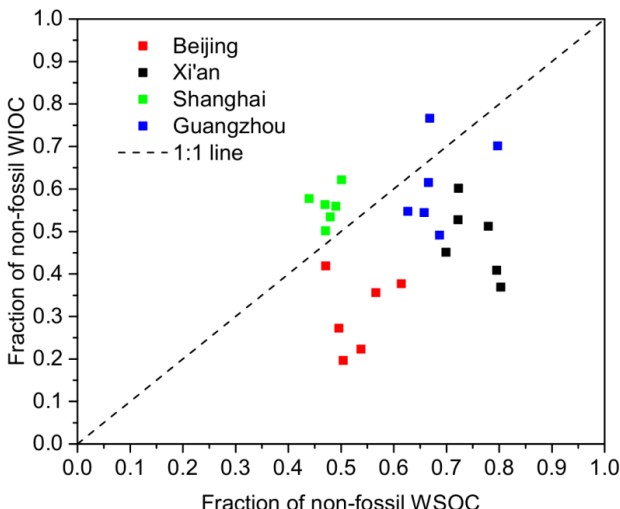


**(a)**

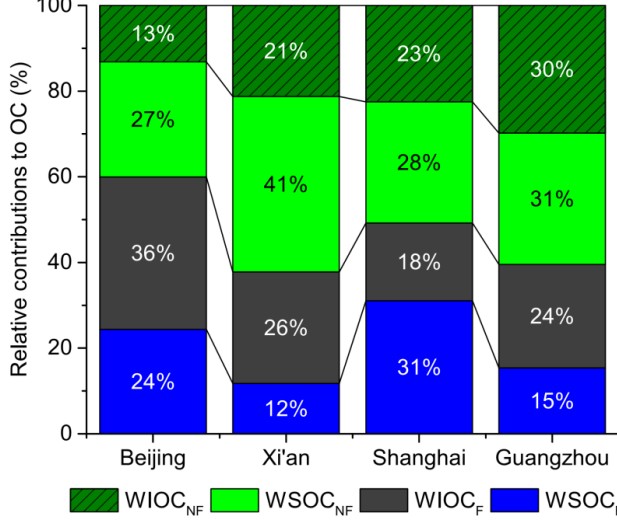


**(b)**





**Figure 6.** Relative contributions (%) to WSOC from biomass burning as well as secondary
organic carbon (SOC) from fossil and non-fossil sources ($WSOC_{SOC,F}$ and $WSOC_{SOC,NF}$,
respectively) in different cities during moderately polluted days (MPD) and heavily polluted
days (HPD) as well as their corresponding excess (Excess=HPD-MPD). The numbers above
the bars refer to the average WSOC concentrations and the SOC fractions (%) of WSOC.

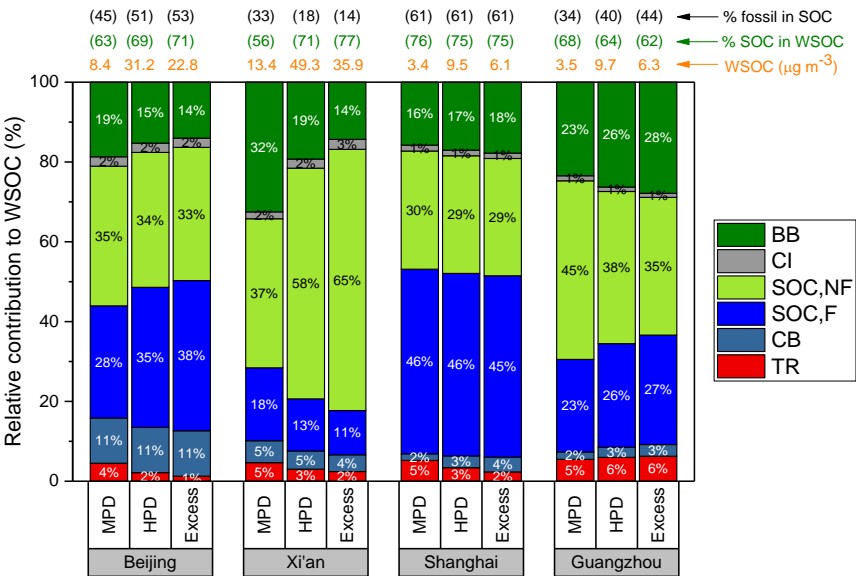






**Figure 7.** Box-plot of the fossil contribution (%) to the WSOC aerosols in East Asia, South
Asia, USA and Europe. The box represents the $25^{th}$ (lower line), $50^{th}$ (middle line) and $75^{th}$ (top
line) percentiles; the empty square within the box represent the mean values; the end lines of
the vertical bars represent the $10^{th}$ (below the box) and $90^{th}$ (above the box) percentiles; the
solid dots represents the maximum and minimum values; the solid diamonds represent the
individual data (Table 1). The data from East Asia is grouped by the winter and non-winter
seasons.

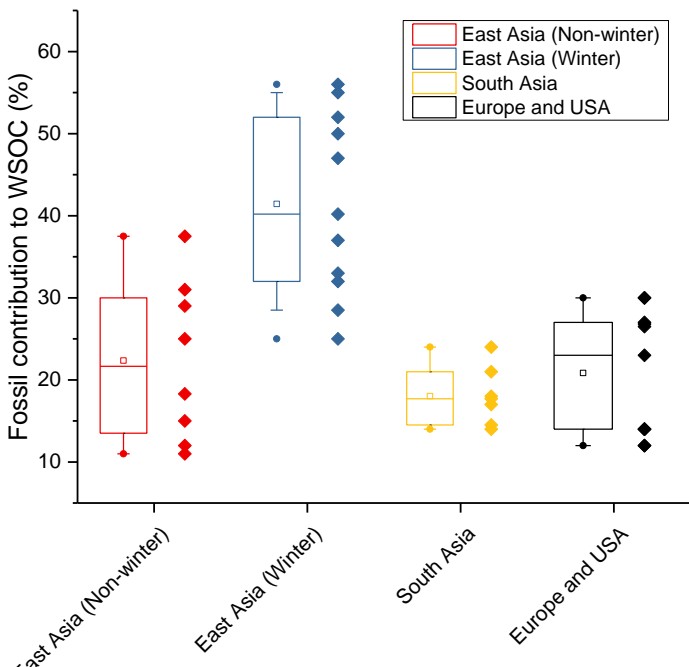
