# Peer review of "Large contribution of fossil-fuel derived secondary organic"

_Atmospheric Chemistry and Physics, 2017_

## Referee Comment (RC1) · Anonymous Referee #1 · 21 Jan 2018

This manuscript combined radiocarbon (14C) and offline-AMS approaches and apportioned sources of organic carbon during an extreme haze episode in China. Here, 14C results were reported for water-soluble OC (WSOC) and water-insoluble OC (WIOC), which enabled a more detailed and straightforward (or accurate) source apportionments of both WSOC and WIOC. Although, radiocarbon measurements of WSOC have been reported in many sites in East Asia and other sites around the world, here the offline-AMS measurements were combined with 14C methods. The fossil and nonfossil sources could be further grouped into primary and secondary sources. Therefore, I think that the method is quite important and may be applied in other regions as well. The results are interesting and informative, which could be applied to some modeling

studies in future. The MS could be published in ACP after the author could address the following minor comments.

Comments: In general, how was the relationship between levogluocosan and non-fossil WSOC (or other OC fractions)?

In the introduction part, I found the authors seems to miss some important references which have reported most recent source apportionment results in winter haze periods in East Asia.

"2.3 Offline-AMS measurement and PMF source apportionment" This part is a bit too long, and I suggest the authors only present the most important part and may cite some papers in any using the same method.

Lines 303-304: what were the major sources of non-fossil emissions in Guangzhou and Xi'an?

---

## Referee Comment (RC2) · Anonymous Referee #2 · 26 Jan 2018

The manuscript "Large contribution of fossil-fuel derived secondary organic carbon to water-soluble organic aerosols in winter haze of China" deals with the source apportionment of water-soluble organic carbon (WSOC). The sources of this carbon fraction are not well known and few studies exist that focus on the source apportionment of WSOC. Therefore this study is of interest and the combination with aerosol mass spectrometer measurements adds very interesting information of primary vs. secondary organic carbon. Overall, I find the manuscript clearly written and the measurements and calculations thorough and accurate.

In my opinion it can be published after relatively minor revisions.

[Figure]

1) Somewhat major comment: The only point that I don't find very clearly described is section 2.3. This section could maybe be shortened on explaining how the PMF works (e.g. Eq 1 could be omitted), but it should contain more detail on the results of the PMF that are relevant for this work.

1a) Are the PMF results from Huang et al. (2014) directly used, or after some modification?

1b) Explain in more detail how the scaling of the factors works in your case. Eg, in line 135 I do not know what you mean by "here only WSOC PMF is present . . .". I suggest not to mention how Huang et al. (2014) scale their factors, because this is only confusing and not relevant for this work.

1c) line 158 – 161, give a bit more detail on these factors (preferably in supporting material)

1c) Eq. 1: you mention sij, but it is not in the equation?

1d) line 161 – 162: "The contribution of the water soluble organic aerosol related to these different factors are extracted . . ." How are they extracted?

1e) line 162 ff: Please provide more detail: What are "the respective OM/OCk ratios" for each factor? Please provide values for each factor and more detail on how they were derived. I think the values should be included in the main text, the rest could be in the supporting material.

Minor comments: 2) Line 191ff: Please give a bit more detail on how you estimate the factor 1.08.

3) Line 236-254: You start the result section with a summary of previous findings of other papers. This would fit better in the introduction

4) line 252: "(three with the highest three with average PM mass)" At first I was confused by this, but I believe that there is just a comma missing?

[Figure]

5) Table 1: Since you have relatively few data points from PM2.5 samples in Europe, I suggest to take a look at a recent publication that also had data related to fossil and non-fossil WSOC. Maybe some useful information can be found in that. Dusek, U., et al., Sources and formation mechanisms of carbonaceous aerosol at a regional background site in the Netherlands: Insights from a year-long radiocarbon study, Atmos. Chem. Phys., 17, 1-19, 2017.

6) Please correct minor grammatical errors throughout the manuscript ... e.g. the example from above: "The contribution of the water soluble organic aerosol related to these different factors are extracted . . .", should either read "The contribution ... IS extracted" or "the contributionS ... are extracted" I noticed several similar instances throughout the manuscript.

---

## Author Comment (AC1) · 16 Feb 2018

We would like to thank the anonymous reviewers and the editor for their careful reading and very important comments. In the following, we replied all the comments point by point.

Anonymous Referee #1

This manuscript combined radiocarbon (14C) and offline-AMS approaches and apportioned sources of organic carbon during an extreme haze episode in China. Here, 14C results were reported for water-soluble OC (WSOC) and water-insoluble OC (WIOC), which enabled a more detailed and straightforward (or accurate) source apportionments of both WSOC and WIOC. Although, radiocarbon measurements of WSOC have been reported in many sites in East Asia and other sites around the world, here the offline-AMS measurements were combined with 14C methods. The fossil and nonfossil sources could be further grouped into primary and secondary sources. Therefore, I think that the method is quite important and may be applied in other regions as well. The results are interesting and informative, which could be applied to some modeling studies in future. The MS could be published in ACP after the author could address the following minor comments.

Reply: We thank the reviewer for the nice summary of our paper and the positive appraisal of the importance of our work. In the following, we replied all the comments point by point.

Comments: In general, how was the relationship between levoglucosan and non-fossil

WSOC (or other OC fractions)?

Reply: We added Figure 4 (see Figure R1 below) in redevised MS. We also added "As shown in Figure 4, non-fossil WSOC was significantly correlated with levoglucosan, indicating that a large fraction of non-fossil WSOC was indeed from biomass burning emissions. In addition, no significant or only a negative correlation (Figure 4) was found between levoglucosan and fraction of fossil to WSOC, suggesting that fossil-fuel source is very unlikely a major or important contributor of levoglucosan even in the regions (e.g., Xi'an and Beijing in this study) where coal combustion is important."

[Figure]

Figure R1. Relationships of non-fossil derived WSOC (WSOC$_{NF}$) and levoglucosan (left), levoglucosan and fraction of fossil to WSOC (f$_F$(WSOC)) (middle) and levoglucosan and fraction of non-fossil to WSOC (f$_{NF}$(WSOC)) (right).

In the introduction part, I found the authors seems to miss some important references which have reported most recent source apportionment results in winter haze periods in East Asia.

Reply: The primary objective of this study is to investigate source of WSOC. So we added one more reference about WSOC source apportionment in China (Zong et al., 2016).

"2.3 Offline-AMS measurement and PMF source apportionment" This part is a bit too long, and I suggest the authors only present the most important part and may cite some papers in any using the same method.

Reply: We still think that the detailed information about offline-AMS and PMF is very important, so we would like to keep in the main text. But we revised this part according to the comments from the reviewer 2 (see details below)

Lines 303-304: what were the major sources of non-fossil emissions in Guangzhou and Xi'an?

Reply: The major sources of non-fossil emissions were biomass burning emissions and SOC formation. The detailed WSOC sources were present in the Sec.3.3 ("High contribution of secondary formation to WSOC") and Figure 7 in revised MS.

Anonymous Referee #2

The manuscript "Large contribution of fossil-fuel derived secondary organic carbon to water-soluble organic aerosols in winter haze of China" deals with the source apportionment of water-soluble organic carbon (WSOC). The sources of this carbon fraction are not well known and few studies exist that focus on the source apportionment of

WSOC. Therefore this study is of interest and the combination with aerosol mass spectrometer measurements adds very interesting information of primary vs. secondary organic carbon. Overall, I find the manuscript clearly written and the measurements and calculations thorough and accurate.

In my opinion it can be published after relatively minor revisions.

Reply: We thank the reviewer for the nice summary of our paper and the positive appraisal of the importance of our work. In the following, we replied all the comments point by point.

1) Somewhat major comment: The only point that I don't find very clearly described is section 2.3. This section could maybe be shortened on explaining how the PMF works (e.g. Eq 1 could be omitted), but it should contain more detail on the results of the PMF that are relevant for this work.

1a) Are the PMF results from Huang et al. (2014) directly used, or after some modification? 1b) Explain in more detail how the scaling of the factors works in your case. Eg, in line 135 I do not know what you mean by "here only WSOC PMF is present . . .". I suggest not to mention how Huang et al. (2014) scale their factors, because this is only confusing and not relevant for this work.

Reply: We agree with the reviewer and we have modified the text accordingly. The PMF results from Huang et al. (2014), but only data relative to WSOC are used.

The modifications include the following:

Line 120, we added: Here, only data relative to WSOC are used.

Line 128, we modified the text by removing the explanation about the methodology followed by Huang. The new text reads as follows:

Online AMS measurements provide quantitative mass spectra of submicron non-refractory aerosol species, including organic aerosol and ammonium nitrate and sulfate. However, the offline AMS measurements described herein cannot be directly related to ambient concentrations due to uncertainties in nebulization and AMS lens cut-off. Here, we have scaled the organic aerosol mass spectra to water soluble organic aerosol concentrations (WSOM), obtained as WSOC times OM/OC ratios. The latter were determined by the high resolution analysis of the organic aerosol mass spectra, acquired by the AMS.

1c) line 158 – 161, give a bit more detail on these factors (preferably in supporting material)

R: We believe that sufficient details were presented in Huang et al. (2014). Here, we will provide only specific details about the identification of these factors. The updated text reads as follows:

Reply: The elements that were constrained in $F_{k,j}$ matrix can be found in Huang et al. (2014). The factors extracted by ME-2 were interpreted to be related to primary emissions from traffic (TR), biomass burning (BB), coal burning (CC), cooking emissions (CI) and dust and from two secondary aerosol fractions. The elements of TR and CI were constrained in the model. BB was identified based on the high contribution of potassium, anhydrous sugars and the fragment $C_2H_4O_2^+$ in the WSOA mass spectrum resulting from the decomposition of the anhydrous sugars. CC, dominant in Beijing, was identified based on the high contribution of PAHs and unsaturated hydrocarbon fragments in the associated WSOA mass spectrum. Dust is identified based on the high contribution of crustal material and calcium. Finally, two secondary mass factors were identified based on the high contribution of secondary inorganic aerosols, including ammonium sulfate and nitratein their factor profiles and the high oxygenation degree of the associated WSOA.

1c) Eq. 1: you mention sij, but it is not in the equation?

Reply: Sij, the measurement uncertainty matrix, is not in Eq.1. We were sorry that we made a mistake in the previous main text. The section was revised. PMF uses the uncertainty matrix to scale the residual matrix $E_{i,j}$, and the ratio of quadratic sum of this ratio is minimized iteratively when Equation 1 is solved.

We have updated this section as follows:

PMF solves the bilinear matrix equation:

$$X_{ij} = \sum_k G_{i,k} F_{k,j} + E_{i,j} \qquad\qquad \text{(Eq. 1)}$$

by following a weighted least squares approach. In the equation, $i$ represent the time index, $j$ a species and $k$ the factor number. $X_{ij}$ is the input matrix, $G_{i,k}$ is the matrix of the factor time-series, $F_{k,j}$ is the matrix of the factor profiles and $E_{i,j}$ the model residual matrix. PMF determines $G_{i,k}$ and $F_{k,j}$ such that the ratio of the Frobenius norm of $E_{i,j}$ over the uncertainty matrix, $s_{i,j}$, used as model input is minimised.

1d) line 161 – 162: "The contribution of the water soluble organic aerosol related to these different factors are extracted . . ." How are they extracted?

Reply: The contributions of the water soluble organic aerosol related to these different factors were determined by the multiplying their relative abundance in the factor profiles by the respective factor time-series. The factors WSOM time series were then divided by the respective $OM/OC_k$ calculated from the high-resolution analysis of the factor mass spectral profile to obtain the $WSOC_k$ time series related to each of the factors.

This had been added to the text.

1e) line 162 ff: Please provide more detail: What are "the respective OM/OCk ratios" for each factor? Please provide values for each factor and more detail on how they were derived. I think the values should be included in the main text, the rest could be in the supporting material.

Reply: The OM/OCk values were presented in Huang et al., 2014. They were determined from the elemental analysis of the organic fragments in the mass spectra. This is a very common procedure used by all the AMS community and we do not think that it is necessary to provide more related details. However, we did add the OM/OCk values in the main text:

The contributions of the water soluble organic aerosol related to these different factors were determined by the multiplying their relative abundance in the factor profiles by the respective factor time-series. The factors WSOM time series were then divided by the respective OM/OC$_k$ calculated from the high-resolution analysis of the factor mass spectral profile to obtain the WSOC$_k$ time series related to each of the factors. The average OM/OC$_k$ are: 1.25, 1.39, 1.49, 1.55, 2.25, and 2.4 for TR, CI, BB, CB, SOA, and dust, respectively.

Minor comments: 2) Line 191ff: Please give a bit more detail on how you estimate the factor 1.08.

Reply: The detailed calculation is explained in (Zhang et al., 2012). The reference is added now. See " $f_{NF}$(ref) is a reference value representing $f_M$ of non-fossil sources during the sampling periods, which can be further separated into biogenic (bio) and biomass-burning (bb) sources given that other non-fossil sources (e.g. cooking and biofuel combustion) are negligible. Hence, $f_{NF}$(ref) is defined as:

$$f_{NF}(ref) = p_{bio} \times f_{bio}(ref) + (1 - p_{bio}) \times f_{bb}(ref)$$

where $p_{bio}$ refers to the percent of the biogenic sources to the total non-fossil sources; $f_{bb}$(ref) can be retrieved from a tree-growth model according to (Mohn et al., 2008), and $f_{bio}$(ref) from the long-term time series of $^{14}CO_2$ measurements in atmosphere at the Schauinsland station (Levin et al., 2010). In the case of source apportionment of OC, $p_{bio}$ can be simply estimated as a constant value (e.g. 50%) given that the variations of $f_{NF}$(ref) produced by different $p_{bio}$ values are relatively small, especially if compared to the measurement and method uncertainties (Minguillón et al., 2011). And in the case of EC, $p_{bio}$ is zero as biomass burning is the only source of non-fossil EC." Here, f$_{NF}$(ref)=0.5*1.1+0.5*1.05, approaching 1.08. We changed the sentence as "$f_{M,ref}$ is a reference value of $f_M$ for non-fossil carbon sources including biogenic and biomass burning emissions, which were estimated as $1.08\pm0.05$ (i.e., $f_{M,ref}=(0.5*1.10+0.5*1.05)$ (see details in (Zhang et al., 2012)) for WSOC samples collected in 2013 according to the contemporary atmospheric $CO_2$ $f_M$ (Levin et al., 2010) and a tree growth model (Mohn et al., 2008).".

3) Line 236-254: You start the result section with a summary of previous findings of other papers. This would fit better in the introduction

Reply: this section is aiming to make comparisons between our study and other studies to show a board implication, so we would like to keep the discussion here.

4) line 252: "(three with the highest three with average PM mass)" At first I was confused by this, but I believe that there is just a comma missing?

Reply: A comma was added.

5) Table 1: Since you have relatively few data points from PM2.5 samples in Europe, I suggest to take a look at a recent publication that also had data related to fossil and non-fossil WSOC. Maybe some useful information can be found in that. Dusek, U., et al., Sources and formation mechanisms of carbonaceous aerosol at a regional background site in the Netherlands: Insights from a year-long radiocarbon study, Atmos. Chem. Phys., 17, 1-19, 2017.

Reply: Thanks! We added it.

6) Please correct minor grammatical errors throughout the manuscript ... e.g. the example from above: "The contribution of the water soluble organic aerosol related to these different factors are extracted . . .", should either read "The contribution ... IS extracted" or "the contributionS ... are extracted" I noticed several similar instances throughout the manuscript.

Reply: Yes. We checked grammatical errors throughout the manuscript

References

[revised manuscript text omitted]